# Generalized Correntropy Criterion-Based Performance Assessment for Non-Gaussian Stochastic Systems

**DOI:** 10.3390/e23060764

**Published:** 2021-06-17

**Authors:** Jinfang Zhang, Guodou Huang, Li Zhang

**Affiliations:** School of Control and Computer Engineering, North China Electric Power University, Beijing 102206, China; 120192227115@ncepu.edu.cn (G.H.); 120192227034@ncepu.edu.cn (L.Z.)

**Keywords:** performance assessment indicators, hybrid-EDA, non-Gaussian systems, generalized correntropy criterion

## Abstract

Control loop performance assessment (CPA) is essential in the operation of industrial systems. In this paper, the shortcomings of existing performance assessment methods and indicators are summarized firstly, and a novel evaluation method based on generalized correntropy criterion (GCC) is proposed to evaluate the performance of non-Gaussian stochastic systems. This criterion could characterize the statistical properties of non-Gaussian random variables more fully, so it can be directly used as the assessment index. When the expected output of the given system is unknown, generalized correntropy is used to describe the similarity of two random variables in the joint space neighborhood controlled and take it as the criterion function of the identification algorithms. To estimate the performance benchmark more quickly and accurately, a hybrid-EDA (H-EDA) combined with the idea of “wading across the stream algorithm” is proposed to obtain the system parameters and disturbance noise PDF. Through the simulation of a single loop feedback control system under different noise disturbances, the effectiveness of the improved algorithm and new indexes are verified.

## 1. Introduction

With the continuous improvement of automation in the industrial process, the production process is becoming more and more complex. Control loops play the most important role in automation systems. Product quality, operation safety, material consumption and energy consumption are closely related to the performance of control system. Excellent control loop performance will ensure the effectiveness of the control systems, thus ensuring product quality and reducing product cost. The significance of performance assessment is to realize, restore and maintain the optimal performance of the control loops at all times. Therefore, it is of great practical value to evaluate the system performance quickly and accurately in industrial operation. At present, control loop performance assessment (CPA) has become an essential technology to ensure the smooth progress of industrial production [1]. 

Noise is unavoidable during the performance evaluation [2]. Most of the current performance evaluation methods assume that the noise obeys the Gaussian distribution. The research focuses on the first or second-order moments of the target output (i.e., the mean and variance). Harris [3] proposed a CPA index based on minimum variance control (MVC). This method is regarded as a milestone in the performance assessment research, so we named the index after the researcher, Harris. Later researchers have made great progress on this fundamental. The MVC-based performance evaluation index has been applied to various types of control systems, and this index has been improved for different situations [4,5,6]. At present, methods of system performance assessment based on MVC are very mature and has a remarkable application effect when the noise disturbance conforms to Gaussian. However, most of the disturbances in the actual industrial process obey non-Gaussian. The traditional MVC methods are no longer applicable for this case. A more suitable indicator should be chosen to reflect the higher-order statistical characteristics of variables rather than mean and variance in the closed-loop control systems.

For a stochastic system with non-Gaussian noises, the probability density function (PDF) of the output data is approximated by B-spline in the early research direction [7]. A stochastic distributed control algorithm based on minimum entropy control (MEC) was proposed by Zhang and Chu [8]. Information entropy has a more general meaning for any stochastic variable than mean or variance. All higher-order moments (not just second-order) can be minimized by minimizing entropy rather than mean square error. Therefore, the application effect of MEC control strategy is much better than MVC in non-Gaussian systems. On this basis, some significant progress has been made in the field of CPA with non-Gaussian noise [9,10,11,12,13,14]. When the expected output PDF is not given in advance, it is necessary for the CPA methods of the controlled system based on MEC to identify the associated system model effectively and systematically based on closed-loop data. Minimum entropy criterion was introduced into the feedback control system by Jiang [9], and a new performance assessment method came into being. In work [9], the model of the associated system is identified by the estimation of distribution algorithm (EDA) to obtain the optimal parameters, then the mathematical model and disturbance sequence between output and disturbance is obtained. It is verified that the new benchmark is appropriate for both Gaussian and non-Gaussian systems. However, the method given by Jiang [9] is not detailed enough, the specific calculation methods of minimum entropy for discrete and continuous disturbances are not given. Fortunately, an improved minimum entropy calculation method was proposed by Zhou [10,11] to correct the defects of Jiang’s method. In work [10], Zhou and Zhang analyzed the defects of Shannon entropy in CPA process and proposed a new entropy index, rational entropy. The method given in [10] can calculate the theoretical benchmark value. In work [11], the traditional EDA has been improved to identify the system parameters and estimate the noise PDF better and the estimation method of benchmark value is also given. In reference [12], the rational entropy index is used to evaluate a typical cascade control system. However, entropy has the property of translation invariance, that is, the size of entropy is only determined by the shape of its distribution. When the mean of error distribution is not around zero, the wrong assessment result may be caused. For this case, Zhou proposed the performance index of mean-constraint, while a CPA method based on mixture correntropy was proposed by Zhang [13]. The problem of the minimum entropy translation invariance can be solved by adjust the kernel width and weight coefficient of mixture correntropy. Similarly, to avoid the defect of Renyi entropy indicator is insensitive to mean shift, Zhang [14] combined the Renyi entropy benchmark with the mean value to construct a new indicator that can reflect the mean shift of non-Gaussian system. In conclusion, the main breakthrough of the performance assessment method based on MEC is the improvement of the identification algorithm and the selection of some new entropy indicators.

Correntropy is a measure of the generalized similarity of two random variables using information from the statistical properties of higher-order signals [15]. The kernel function of mixture correntropy CPA [13] is Gaussian kernel by default. However, it can only reflect a specific type of noise with Gaussian function, and the shape of entropy cannot be changed freely. Therefore, the Gaussian kernel is not always the best choice [16]. A more general correntropy, generalized correntropy (GC) is proposed. The generalized Gaussian density function is used to replace the traditional Gaussian kernel function for correlation analysis. With greater flexibility, generalized correntropy could describe the statistical characteristics of non-Gaussian random variables better and more fully. In recent years, generalized correntropy has been widely used to design the robust adaptive filters to adapt to different applications [17,18,19,20]. For instance, to improve the accuracy and stability of the traditional multi-kernel filtering algorithms, the multi-kernel generalized maximum correntropy (MKGMC) filtering algorithm is developed by Sun [17] under the generalized maximum correntropy criterion. Wang [18] applied generalized correntropy to the design of sparse Gaussian Hermite orthogonal filters. Qiu [19] replaced the mean square error loss with generalized correntropy loss (GCL) in the original unscented Kalman filter (UKF) framework, which combines the strength of GCL developed in robust information theoretic learning to solve the non-Gaussian interference and the strength of the UKF in handing strong model nonlinearities. Therefore, it is very promising to study the performance assessment of non-Gaussian stochastic systems by replacing entropy criterion with generalized correntropy.

To estimate the performance benchmark better, this paper integrates the idea of wading across the stream algorithm (WSA) into the traditional EDA to improve the local convergence ability of the algorithm. In the iterative process of the algorithm, the center of some excellent individuals is selected to cross with the best solution to fully retain the information of the best individuals. Generalized correntropy is used as the fitness value of H-EDA to identify the parameters of the controlled auto regression and moving average (CARMA) model of the controlled system, which accelerate the algorithm and improves the accuracy. The effectiveness of the hybrid algorithm is verified through the simulation of a single loop feedback control system under different noise disturbances.

In the next section, the CPA index based on MEC is reviewed, and the generalized correntropy index is proposed to overcome the shortcomings of the previous entropy indexes. The third part introduces the performance evaluation process based on the H-EDA in detail. In the fourth section, a detailed simulation of a single input single output (SISO) system is carried out, and the final conclusion will be given in the last part.

## 2. Performance Assessment Based on MEC 

### 2.1. Feedback Control Loop

A SISO feedback control system is considered in this paper for convenience of research, the detailed description is given in Figure 1.

where, r represents the system set value; u represents the output of the controller; v represents stationary independent identically distributed unmeasurable noise.

The system in Figure 1 could be described as a CARMA model as follows,
(1)$$A(z^{-1})y(k)=z^{-\tau }B(z^{-1})u(k)+C(z^{-1})v(k)$$
where y(k), u(k) and v(k) are the output, input and unknown noise disturbance of CARMA model process respectively,
(2)$$\left\{ \begin{array}{l} A(z^{-1})=1+a_{1}z^{-1}+a_{2}z^{-2}+\cdot \cdot \cdot +a_{na}z^{-na}\\ B(z^{-1})=b_{1}z^{-1}+b_{2}z^{-2}+\cdot \cdot \cdot +b_{nb}z^{-nb}\\ C(z^{-1})=1+c_{1}z^{-1}+c_{2}z^{-2}+\cdot \cdot \cdot +c_{nc}z^{-nc} \end{array}\right.$$
where $n_{a}$, $n_{b}$, $n_{c}$ and τ are the structural parameters of the model; $n_{a}$, $n_{b}$, and $n_{c}$ are the order of ${A(z}^{-1})$, ${B(z}^{-1})$ and ${C(z}^{-1})$ respectively, τ is the system delay. The parameters $a_{na}$, $b_{nb}$, and $c_{nc}$ can be estimated by the model identification, provided the structural parameters are obtained.

### 2.2. Minimum Entropy Index

Assuming that the input of the given system is 0, the output is,
(3)$$y_{t}=\frac{G_{v}}{1+G_{p}G_{c}}v_{t}=\frac{G_{v}}{1+z^{-\tau }{\tilde{G}}_{p}G_{c}}v_{t}$$
where, $\tilde{G}p$ is the transfer function without delay. By solving the Diophantine equation, the disturbance transfer function
$G_{v}$ could be further decomposed,
(4)$$G_{v}(z^{-1})=F(z^{-1})+z^{-\tau }R(z^{-1})=(1+n_{1}z^{-1}+n_{2}z^{-2}+\cdot \cdot \cdot +n_{\tau -1}z^{-(\tau -1)})+z^{-\tau }R(z^{-1})$$
where, ${F(z}^{-1})$ is the impulse response coefficient of the disturbance transfer function $G_{v}$ and ${R(z}^{-1})$ is the remaining transfer function satisfying the identity (4),
(5)$$y(t)=Fv(t)+Lv(t-\tau )=\underset{\mathrm{feedback}-\mathrm{invariant}}{\underbrace{\left(n_{0}+n_{1}z^{-1}+n_{2}z^{-2}+\cdot \cdot \cdot +n_{v}z^{-(\tau -1)})v(t\right)}}+\underset{\mathrm{feedback}-\mathrm{varying}}{\underbrace{\left(n_{d}z^{-\tau }+n_{d+1}z^{-(\tau +1)}+\cdot \cdot \cdot )v(t\right)}}$$
where,
(6)$$L=\frac{R-F{\tilde{G}}_{p}G_{c}}{1+z^{-\tau }{\tilde{G}}_{p}G_{c}}$$

The control objective of the minimum entropy controller can be achieved by minimizing the entropy of the output variables when analyzing the linear non-Gaussian system [21,22]. The feedback-invariant only depend on the characteristics of the disturbance in the process rather than the function of the process model or controller. The second term on the right of Equation (5) is the feedback-varying, which means the controller $G_{v}$ can determine the entropy value of the process output. Since both sides of Equation (6) are independent, therefore,
(7)$$H(y_{t})=H(Fv_{t}+Lv_{t-\tau })\geq H(Fv_{t})$$

Just like the MVC method, the equal sign in Equation (7) holds only when L=0, then the output entropy reaches the minimum, that is the benchmark entropy,
(8)$$H^{\min}(y_{t})=H(Fv_{t})$$

The CPA based on MEC is to compare the actual system output entropy ${H(y}_{t})$ with the output entropy $H^{\min}{(y}_{t})$ under MEC. To sum up, the CPA index based on MEC can be expressed as,
(9)$$\eta =\frac{H^{\min}(y_{t})}{H(y_{t})}=\frac{H(Fv_{t})}{H(y_{t})}$$

This indicator is similar to MVC, which is always between 0 and 1. Generally, the closer it is to 1, the closer the system to the ideal case, indicating that the system performs better; conversely, the closer it is to 0, the worse the system performance is, even including unstable control.

### 2.3. Prevenient Entropy Index and Generalized Correntropy 

Entropy is a measure to describe the uncertainty of random variables, even for random variables without mean or variance. The size of entropy is only determined by the shape of the distribution rather than its location. There are many ways to describe entropy, such as Shannon entropy (*SE*), rational entropy (*RE*), Renyi entropy and delta entropy. In the researches based on the minimum entropy criterion, *SE* is widely used. For linear Gaussian systems, *SE* is equivalent to variance. For a continuous random variable x, Shannon entropy can be expressed as,
(10)$$H_{SE}=-\int \gamma (x)\ln\gamma (x)dx,x\in R$$
where γ(x) is its probability density function (PDF). *SE* is one of the CPA criteria based on MEC system in previous researches [10]. However, it does not meet the “consistency” principle, that is, the results must be the same when entropy could be calculated in many different ways. These characteristics show that *SE* cannot be used as a new evaluation index. 

Fortunately, the rational entropy (*RE*) criterion is proposed by Zhou [10], which has most of the properties of Shannon entropy and satisfies the “consistency” principle. The definition of rational entropy is,
(11)$$H_{RE}=-\int \gamma (x)\log\frac{\gamma (x)}{1+\gamma (x)}dx,x\in R$$
where R is the domain of random variable x, and γ(x) is its PDF. *RE* is a suitable benchmark for CPA of linear non-Gaussian systems. By comparing the output rational entropy under MEC process with actual, a CPA index will be obtained.

Of course, *RE* is not perfect. The PDF of variable x is essential to calculate the output *RE*. If the PDF is known, it is more convenient to use the *RE* criterion to calculate the CPA index. However, when the PDF of the variable is unknown, this method is very complicated in the actual calculation process. For instance, when the system parameters are estimated by the EDA, *RE* as the fitness value, needs to be calculated for several times in each iteration (usually 1000 s), and the operation will be repeated for many times in the identification process. In the actual systems, the parameters and data will be more complex, this method will waste a lot of time. Therefore, this method lacks practical engineering significance.

The mixture correntropy index is adopted by Zhang [13] to solve the problem of translation invariance. However, the Gaussian kernel is applied to the kernel function of mixture correntropy in [13], which is not comprehensive in describing the statistical properties of non-Gaussian random variables [16]. As mentioned before, generalized correntropy is a more appropriate indicator.

Correntropy is a measure of local similarity, which is directly related to the similarity of two random variables in the joint space neighborhood controlled by kernel width. Given two random variables X and Y, their correntropy is defined as [16],
(12)$$V_{\sigma }(X,Y)=E[\kappa_{\sigma }(X,Y)]=\int \kappa_{\sigma }(x,y)dF_{XY}(x,y)$$
where E[⋅] is the expectation value, $\kappa_{\sigma }$ is the kernel function with the width of σ(σ>0) and $F_{\mathrm{XY}}\left(x,\;y\right)$ denotes the joint distribution function of (X, Y). Generally, Gaussian kernel is chosen as the kernel function of correntropy,
(13)$$\kappa_{\sigma }(x,y)=G_{\sigma }(e)=\frac{1}{\sqrt{2\pi }\sigma }\exp\left(-\frac{e^{2}}{2\sigma^{2}}\right)=\frac{1}{\sqrt{2\pi }\sigma }\exp\left(-\lambda e^{2}\right)$$
where e=x−y and ${\lambda =1/2\sigma }^{2}$ is the kernel parameter. In this paper, a well-known GGD function will be used as the kernel function in correntropy [23],
(14)$$G_{\alpha ,\beta }(e)=\frac{1}{2\beta \Gamma (1/\alpha )}\exp\left(-{\left|\frac{e}{\beta }\right|}^{\alpha }\right)=\gamma_{\alpha ,\beta }\exp\left(-\lambda {\left|e\right|}^{\alpha }\right)$$
where Γ(⋅) means the gamma function, α>0 represents shape parameter; β>0 is bandwidth parameter, ${\lambda =1/\;\beta \;}^{\alpha }$ is the kernel parameter and $\gamma_{\alpha ,\;\beta }=\alpha /(2\beta \Gamma (1/a))$ is the normalization constant.

The GGD function is used as the kernel function in the correntropy,
(15)$$V_{\alpha ,\beta }(X,Y)=E[G_{\alpha ,\beta }(E_{X-Y})]=E[G_{\alpha ,\beta }(X-Y)]$$
where, to distinguish the correntropy with Gaussian kernel, we define it as generalized correntropy (GC). Clearly, it is an extension of correntropy. 

If the joint distribution f (x, y) of X and Y is known, the correntropy could be calculated by the following formula,
(16)$$V_{\alpha ,\beta }(X,Y)=\int_{-\infty }^{+\infty }\int_{-\infty }^{+\infty }G_{\alpha ,\beta }(x-y)f(x,y)dxdy$$

Generally, the joint distribution of X and Y is hard to known. Similar to correntropy, generalized correntropy is also calculated by sample estimation,
(17)$${\overset{\wedge }{V}}_{N,\alpha ,\beta }(X,Y)=\frac{1}{N}\sum_{i=1}^{N}G_{\alpha ,\beta }(x(i)-y(i))=\frac{1}{N}\sum_{i=1}^{N}G_{\alpha ,\beta }(e(i))$$

Next, some properties of generalized correntropy are summarized as follows to explain the internal conditions for it to be used as a performance index:

Property 1: $V_{\alpha ,\;\beta }\left(X,\;Y\right)$ is symmetric, that is $V_{\alpha ,\;\beta }{(X,\;Y)=V}_{\alpha ,\;\beta }\left(X,\;Y\right)$;

Property 2:$\;V_{\alpha ,\;\beta \;}\left(X,\;Y\right)$ is positive and bounded: ${0<V}_{N,\;\alpha ,\;\beta }(X,\;Y)\;\leq {\;G}_{\alpha ,\;\beta }{(0)=\gamma }_{\alpha ,\;\beta }$, which reaches the maximum value if and only if X=Y;

Property 3: The generalized correntropy contains the absolute higher moments of error variables, $E_{X-Y}=X-Y$, $V_{\alpha ,\beta }(X,Y)=\gamma_{\alpha ,\beta }\sum_{n=0}^{\infty }\frac{{\left(-\lambda \right)}^{n}}{n!}E\left[{\left|X-Y\right|}^{\alpha n}\right]$.

These characteristics show that generalized correntropy can be applied to CPA. When the expected output of the given system is known, the target PDF is expected to be tracked by the controlled output PDF as much as possible, that is, the expected output $R_{k}$ and the actual output $Y_{k}$ are as consistent as possible, $e_{k}{=R}_{k\;}{-\;Y}_{k}$ is close to 0 infinitely. Ideally, the target PDF is tracked by the output completely, the performance of the control system reaches the optimum, and the generalized correntropy reaches the maximum value $\gamma_{\alpha ,\;\beta }$. Therefore, the new CPA index based on generalized correntropy can be defined as,(18)$$\eta_{GC}\;=\frac{V_{\alpha \beta }(R_{k}-Y_{k})}{\gamma_{\alpha ,\beta }}=\frac{{E\;[G}_{\alpha ,\;\beta }{(e}_{k})]}{\gamma_{\alpha ,\beta }}$$
where: ${E\;[G}_{\alpha ,\;\beta }{(e}_{k})]$ is the generalized correntropy of $e_{k}{=R}_{k\;}{-\;Y}_{k}$. Obviously, $0\;\leq {\;\eta }_{GC}\;\leq \;1$, closer the index is to 1, the better the control system performance is; otherwise, the system performance is poor and needs to be improved.

When the expected output of the system is unknown, it is essential to identify the associated systems. In this case, generalized correntropy could be combined with the EDA to generate a new identification algorithm.

In data analysis of regression and classification, a measure called the correntropic loss (C-loss) is always used instead of correntropy [24]. The GCL between *X* and *Y* could be defined as,
(19)$$\begin{array}{ll} J_{GC-loss}(X,Y) & =G_{\alpha ,\beta }(0)-V_{\alpha ,\beta }(X,Y)\\ & =\gamma_{\alpha ,\beta }-V_{\alpha ,\beta }(X,Y) \end{array}$$

Obviously, minimizing GCL corresponds to maximizing generalized correntropy.

Assuming that ${\left\{\left(x_{i},y_{i}\right)\right\}}_{i=1}^{N}$ is a sample taken from $p_{XY}$, the estimated value of GCL is,
(20)$$\begin{array}{ll} {\overset{\wedge }{J}}_{GC-loss}(X,Y) & =G_{\alpha ,\beta }(0)-{\overset{\wedge }{V}}_{\alpha ,\beta }(X,Y)\\ & =\gamma_{\alpha ,\beta }-\frac{1}{N}\sum_{i=1}^{N}G_{\alpha ,\beta }(x_{i}-y_{i})\;\\ & =\gamma_{\alpha ,\beta }-\frac{1}{N}\sum_{i=1}^{N}G_{\alpha ,\beta }(e_{i}) \end{array}$$

GCL could be applied as the fitness value of the H-EDA to search for the optimal parameters and estimate the noise PDF of the given system; the specific algorithm will be given in Section 3.

## 3. Improvement and Application of EDA in CPA 

In essence, the problem of system parameter identification is an alternative problem of high-dimensional parameter space optimization, so the system parameters can be obtained by the methods of high-dimensional parameters optimization. To obtain the noise PDFs, we have to know the order, time delay and the parameters of the given system.

For the estimation of delay, the simplest and most commonly used method is to analyze the correlation between the input $u_{t}$ and output $y_{t}$ signals [25], which can be expressed as,(21)$${\hat{\tau }}_{d}\;=\underset{\tau }{\;max}{E[y(\tau )u(\tau -\tau }_{d})]$$

When ${\tau =\tau }_{d}$, $R_{y,\;u}\left(\tau \right)$ reaches the peak, the corresponding time is the estimated value of delay.

The Akaike information criterion [26] is applied to get the order of the system in this paper. For the given CARMA model, AIC criterion is as follows, $n_{c}$ is the order of the noise model,
(22)$$AIC(n)=L\mathrm{lg}\sigma_{e}^{2}+a(n_{a}+n_{b}+n_{c})$$

### 3.1. EDA and Improved

Estimation of distribution algorithm (EDA) is a randomized search heuristic algorithm, which is used to create a probabilistic model of solution space. This model is updated iteratively based on the quality of the solutions sampled by the model [27]. In recent years, EDA as an optimization method has received great attention, and applied to scheduling, project, machine learning and identification problems successfully [27,28,29,30]. Wang [28] proposed a hybrid algorithm named estimation of particle swarm distribution algorithm (EPSDA), which combines PSO (the local search method) with the EDA (the global search method) to improve the efficiency of solving nonlinear bilevel programming problems (NBLP). To overcome the instability of model updating in the iterative process, a novel EDA based on the classic compact genetic algorithm (cGA) is proposed by [27] to optimize the benchmark function ONEMAX. Liang [29] developed a new variant of Gaussian EDA (GEDA) to solve the problem of premature convergence in traditional GEDA. In the process of CPA, EDA is mainly devoted to estimate the parameters of the given systems. Through appropriate improvements, the given system can be identified faster and more accurately, so as to obtain a more accurate benchmark.

### 3.2. Parameter Identification Based on Hybrid-EDA

As the previous work shows, the traditional EDA still has some shortcomings in the optimization process, such as a large amount of calculation, poor accuracy, etc. Therefore, based on the traditional EDA, the space for parameter identification is first obtained through preliminary estimation. To improve the local convergence ability, incorporating WSA [31] ideas into the EDA. In the iterative process, the centers of some excellent individuals are selected to cross-operate with the best individuals, the information of outstanding individuals could be retained more fully.

#### 3.2.1. Acquisition of Parameter Identification Space

The initialization of parameter space is of positive significance to improve the identification accuracy and shorten the time required, so it is necessary to estimate the model parameters preliminarily. The recursive extended least squares (RELS) algorithm is selected in this paper. 

The above CARMA model can be expressed as,
(23)$$y(k)=\varphi^{T}(k)\theta +v(k)$$
where,
(24)$$\left\{ \begin{array}{l} \varphi (k)=[-y(k-1),\cdot \cdot \cdot ,-y(k-n_{a}),u(k-\tau ),\cdot \cdot \cdot ,u(k-\tau -n_{b}),v(k-1),\cdot \cdot \cdot ,v(k-n_{c}){]}^{T}\\ \theta ={\left[a_{1},\ldots ,a_{n_{a}},b_{0},\ldots ,b_{n_{b}},c_{1},\ldots ,c_{n_{c}}\right]}^{T}\in R^{(n_{a}+n_{b}+1+n_{c})\times 1} \end{array}\right.$$

Since v(k) is unmeasurable, it can only be replaced by its estimated value $\hat{v}(k)$,
(25)$$\hat{v}(k)=y(k)-\hat{y}(k)=y(k)-{\hat{\varphi }}^{T}(k)\hat{\theta }$$
where,
(26)$$\left\{ \begin{array}{l} \hat{\varphi }(k)=[-y(k-1),\cdot \cdot \cdot ,-y(k-n_{a}),u(k-\tau ),\cdot \cdot \cdot ,u(k-\tau -n_{b}),\hat{v}(k-1),\cdot \cdot \cdot ,\hat{v}(k-n_{c}){]}^{T}\\ \hat{\theta }=[{\hat{a}}_{1},\ldots ,{\hat{a}}_{n_{a}},{\hat{b}}_{0},\ldots ,{\hat{b}}_{n_{b}},{\hat{c}}_{1},\ldots ,{\hat{c}}_{n_{c}}{]}^{T}\in R^{(n_{a}+n_{b}+1+n_{c})\times 1} \end{array}\right.$$

The purpose is to obtain the parameter vector $\hat{\theta }$ when the objective function J$(\hat{\theta })$ reaches the minimum value. $J(\hat{\theta })$ is defined as follows,
(27)$$J(\hat{\theta })=\sum_{k=1}^{L}v^{2}(k)=\sum_{k=1}^{L}[y(k)-\varphi^{T}(k)\hat{\theta }{]}^{2}$$

Based on the estimation of Gaussian system parameters, the initial estimates of variance ${\hat{\sigma }}_{v}$ and parameters $\hat{\theta }$ could be obtained, then the parameter identification space $\Omega_{W}$ could be taken as $\hat{\theta }$ ± 3${\hat{\sigma }}_{v}$.

#### 3.2.2. The Idea of Wading Across Stream Algorithm

Wading across stream algorithm (WSA) [31] is inspired by the idea of “crossing the river with stones”. This algorithm examines the “shore” carefully to select an initial starting point firstly, several solutions are randomly searched out in the neighborhood near the starting point to get the best one as the iterative result. Then take this result as the starting point and continue search for several solutions in the nearby neighborhood, select the optimal solution as the third iterations result, and so on until the termination condition is satisfied. The WSA is similar to the simulated annealing algorithm but the effect is better, and the algorithm idea is simple with good local convergence ability, so we integrate this algorithm idea into the EDA to improve its computational efficiency.

(1)Choose the initial starting point carefully

When the parameter identification space is known, the initially selected solution will have a greater impact on the entire algorithm. Firstly, generate R solutions in the parameter space with uniform random values, and select the optimal solution as the initial. For the parameter vector $\hat{\theta }=[{\hat{a}}_{1},\ldots ,{\hat{a}}_{{}_{na}},{\hat{b}}_{0},\ldots ,{\hat{b}}_{{}_{nb}},{\hat{c}}_{1},\ldots ,{\hat{c}}_{{}_{nc}}]$ to be identified, it is necessary to uniformly and randomly take values in the interval $\left[\hat{\theta }-3{\hat{\sigma }}_{v},\hat{\theta }+3{\hat{\sigma }}_{v}\right]$ to generate R solutions (vectors) to form a parameter space A,
(28)$$A=[{\hat{\theta }}_{1};{\hat{\theta }}_{2};\cdot \cdot \cdot ;{\hat{\theta }}_{R}]=\left[\begin{array}{ccccccccc} {\hat{a}}_{11} & \cdot \cdot \cdot & {\hat{a}}_{1na} & {\hat{b}}_{10} & \cdot \cdot \cdot & {\hat{b}}_{1nb} & {\hat{c}}_{11} & \cdot \cdot \cdot & {\hat{c}}_{1nc}\\ {\hat{a}}_{21} & \cdot \cdot \cdot & {\hat{a}}_{2na} & {\hat{b}}_{20} & \cdot \cdot \cdot & {\hat{b}}_{2nb} & {\hat{c}}_{21} & \cdots & {\hat{c}}_{2nc}\\ \vdots & \vdots & \vdots & \vdots & \vdots & \vdots & \vdots & \ddots & \vdots \\ {\hat{a}}_{R1} & \cdot \cdot \cdot & {\hat{a}}_{Rna} & {\hat{b}}_{R0} & \cdot \cdot \cdot & {\hat{b}}_{Rnb} & {\hat{c}}_{R1} & \cdot \cdot \cdot & {\hat{c}}_{Rnc} \end{array}\right]$$

To calculate the fitness value of these R solutions, select the optimal individual ${\hat{\theta }}^{*}$ as the initial starting point according to the fitness value. For ease of description, denote the starting point is as ${\hat{\theta }}^{*}{=[x}_{1}{}^{*},\;x_{2}{}^{*},\cdot \cdot \cdot {,x}_{n}{}^{*}\;]$, where n=na+nb+nc+1.

(2)Search strategy 

Due to the complexity of parameter identification in high-dimensional space, the design of an effective local search strategy could improve the quality of the solution significantly. According to the idea of “crossing the river with stones”, when you touch a “stone”, you must take that “stone” as the starting point to explore other “stones” around it. Similarly, search for *m* individuals in the neighborhood radius $L_{k}$ around the starting point ${\hat{\theta }}^{*}$ to form a new parameters space B,
(29)$$\begin{array}{l} B=[\hat{\theta }{{}^{\prime }}_{1};\hat{\theta }{{}^{\prime }}_{2};\cdot \cdot \cdot ;\hat{\theta }{{}^{\prime }}_{m}]\\ =\left[\begin{array}{ccccccccc} \hat{a}{{}^{\prime }}_{11} & \cdot \cdot \cdot & \hat{a}{{}^{\prime }}_{1na} & \hat{b}{{}^{\prime }}_{10} & \cdot \cdot \cdot & \hat{b}{{}^{\prime }}_{1nb} & \hat{c}{{}^{\prime }}_{11} & \cdot \cdot \cdot & \hat{c}{{}^{\prime }}_{1nc}\\ \hat{a}{{}^{\prime }}_{21} & \cdot \cdot \cdot & \hat{a}{{}^{\prime }}_{2na} & \hat{b}{{}^{\prime }}_{20} & \cdot \cdot \cdot & \hat{b}{{}^{\prime }}_{2nb} & \hat{c}{{}^{\prime }}_{21} & \cdots & \hat{c}{{}^{\prime }}_{2nc}\\ \vdots & \vdots & \vdots & \vdots & \vdots & \vdots & \vdots & \ddots & \vdots \\ \hat{a}{{}^{\prime }}_{m1} & \cdot \cdot \cdot & \hat{a}{{}^{\prime }}_{mna} & \hat{b}{{}^{\prime }}_{m0} & \cdot \cdot \cdot & \hat{b}{{}^{\prime }}_{mnb} & \hat{c}{{}^{\prime }}_{m1} & \cdot \cdot \cdot & \hat{c}{{}^{\prime }}_{mnc} \end{array}\right]=\left[\begin{array}{cccc} x{{}^{\prime }}_{11} & x{{}^{\prime }}_{12} & \cdots & x{{}^{\prime }}_{1n}\\ x{{}^{\prime }}_{21} & x{{}^{\prime }}_{22} & \cdots & x{{}^{\prime }}_{2n}\\ \vdots & \vdots & \ddots & \vdots \\ x{{}^{\prime }}_{m1} & x{{}^{\prime }}_{m2} & \cdots & x{{}^{\prime }}_{mn} \end{array}\right] \end{array}$$
where ${x^{{}^{\prime }}}_{ij}{=x}_{j}{}^{*\;}{+L}_{k}\cdot r_{ij}$, $r_{ij}$ represents uniformly distributed random numbers in the interval [–1, 1]. Generally, the initial value $L_{0}$ of neighborhood radius is taken as one tenth of the whole parameters space, that is $L_{0}=(0.1\sim 0.3){\hat{\sigma }}_{v}$. To accelerate the later convergence of the algorithm, reduce the neighborhood radius gradually with the iterative process. The simplest method is to set $L_{k}{=\alpha L}_{k-1}$, α is the shrinkage coefficient, generally [0.90, 0.99]. The latest optimal individual ${\hat{\theta }}^{*}$ can be obtained by calculating the fitness values of these m solutions.

#### 3.2.3. Crossover Operation

The optimal individual obtained by traditional identification algorithm only contains the information of one individual, which ignores the information contained in other excellent individuals to a certain extent. The improved method is crossover operation, sort the above m individuals according to the fitness value, select the top $N^{*}$ ($N{}^{*}<m$) excellent solutions (individuals with the best fitness value) *D*,
(30)$$\begin{array}{ll} D & =[\hat{\theta }{}^{\prime }{{}^{\prime }}_{1};\hat{\theta }{}^{\prime }{{}^{\prime }}_{2};\cdot \cdot \cdot ;\hat{\theta }{}^{\prime }{{}^{\prime }}_{N{}^{*}}]\\ & \;=\left[\begin{array}{ccccccccc} \hat{a}{}^{\prime }{{}^{\prime }}_{11} & \cdot \cdot \cdot & \hat{a}{}^{\prime }{{}^{\prime }}_{1na} & \hat{b}{}^{\prime }{{}^{\prime }}_{10} & \cdot \cdot \cdot & \hat{b}{}^{\prime }{{}^{\prime }}_{1nb} & \hat{c}{}^{\prime }{{}^{\prime }}_{11} & \cdot \cdot \cdot & \hat{c}{}^{\prime }{{}^{\prime }}_{1nc}\\ \hat{a}{}^{\prime }{{}^{\prime }}_{21} & \cdot \cdot \cdot & \hat{a}{}^{\prime }{{}^{\prime }}_{2na} & \hat{b}{}^{\prime }{{}^{\prime }}_{20} & \cdot \cdot \cdot & \hat{b}{}^{\prime }{{}^{\prime }}_{2nb} & \hat{c}{}^{\prime }{{}^{\prime }}_{21} & \cdots & \hat{c}{}^{\prime }{{}^{\prime }}_{2nc}\\ \vdots & \vdots & \vdots & \vdots & \vdots & \vdots & \vdots & \ddots & \vdots \\ \hat{a}{}^{\prime }{{}^{\prime }}_{N{}^{*}1} & \cdot \cdot \cdot & \hat{a}{}^{\prime }{{}^{\prime }}_{N{}^{*}na} & \hat{b}{}^{\prime }{{}^{\prime }}_{N{}^{*}0} & \cdot \cdot \cdot & \hat{b}{}^{\prime }{{}^{\prime }}_{N{}^{*}nb} & \hat{c}{}^{\prime }{{}^{\prime }}_{N{}^{*}1} & \cdot \cdot \cdot & \hat{c}{}^{\prime }{{}^{\prime }}_{N{}^{*}nc} \end{array}\right]\\ & \;=\left[\begin{array}{cccc} x{}^{\prime }{{}^{\prime }}_{11} & x{}^{\prime }{{}^{\prime }}_{12} & \cdots & x{}^{\prime }{{}^{\prime }}_{1n}\\ x{}^{\prime }{{}^{\prime }}_{21} & x{}^{\prime }{{}^{\prime }}_{22} & \cdots & x{}^{\prime }{{}^{\prime }}_{2n}\\ \vdots & \vdots & \ddots & \vdots \\ x{}^{\prime }{{}^{\prime }}_{N{}^{*}1} & x{}^{\prime }{{}^{\prime }}_{N{}^{*}2} & \cdots & x{}^{\prime }{{}^{\prime }}_{N{}^{*}n} \end{array}\right] \end{array}$$

Calculate the center point of *D*,
(31)$$\left\{ \begin{array}{l} \bar{\theta }={\left[{\bar{x}}_{1},{\bar{x}}_{2},\ldots ,{\bar{x}}_{n}\right]}^{T}\\ {\bar{x}}_{i}=\frac{1}{N{}^{*}}\cdot \sum_{j=1}^{N}x_{ji} \end{array}\right.$$

Then, the center point is crossed with the best solution, and the optimal individual ${\hat{\theta }}^{*}$ is modified as follows,
(32)$${\hat{\theta }}^{*}=a{\hat{\theta }}^{*}+(1-a)\bar{\theta }$$
where: a is a random number between 0 to 1.

Through the crossover operation, the information of excellent individuals could be retained to the maximum extent, which can avoid falling into the local optimum.

#### 3.2.4. Selection of Fitness Value 

From Equation (23), for the CARMA model in this paper, the error at time k could be defined as,
(33)$$e(k)=y(k)-\hat{y}(k)=y(k)-{\hat{\varphi }}^{T}(k)\hat{\theta }$$

For the estimated error sequence $e=[e_{1},e_{2},\ldots e_{L}]$, the GCL could be defined as,
(34)$${\overset{\wedge }{J}}_{GC-loss}(e)=\gamma_{\alpha ,\;\beta }-\frac{1}{N}\sum_{i=1}^{N}G_{\alpha ,\beta }(e)$$

The optimal model parameters and the relationship between output and disturbance could be obtained by minimizing GCL. When the GCL reaches the minimum, the corresponding parameters are also optimal, namely,
(35)$${\hat{\theta }}^{opt}=\arg\underset{\theta \in \Omega_{W}}{\min}J_{GC}(e_{k})$$
where: $\Omega_{W}$ is the parameter identification space.

#### 3.2.5. Algorithm Summary

The following Algorithm 1 is the program steps of the H-EDA identification algorithm.
**Algorithm 1** Program steps for the identification algorithm. 1. Describe the system by CARMA model; estimate the system delay τ by analyzing the correlation between *u*(*t*) and *y*(*t*); determine the model order ($n_{a}$, $n_{b},\;n_{c}$).2. Preliminary estimate. Rough model parameters are obtained by RELS, determine the identification space $\Omega_{W}$ as $\hat{\theta }\pm 3{\hat{\sigma }}_{v}$.3. Choose the initial starting point carefully. Randomly generate R individuals from the parameter space $\Omega_{W}$ ,A=$\left[{\hat{\theta }}_{1};\;{\hat{\theta }}_{2};\ldots ;\;{\hat{\theta }}_{R}\right]$, calculate the fitness value (error entropy) of these solutions in *A* and select the optimal one ${\hat{\theta }}_{0}{}^{*}$ according to the fitness value (corresponding error entropy is minimum). 4. Search for *m* individuals $B_{0}=\left[{\hat{\theta }}_{1};\;{\hat{\theta }}_{2};\ldots ;\;{\hat{\theta }}_{m}\right]$ within the neighborhood radius $L_{0}$ of ${\hat{\theta }}_{0}{}^{*}$; I=1;  **While** I <= nmax(1) Calculate the fitness value (error entropy) of *m* individuals and sorted $C_{I}=$
$\left[{\hat{\theta }}_{1}{}^{(I){}^{*}};\;{\hat{\theta }}_{2}{}^{(I)*}\;;\ldots ;{\hat{\;\theta }}_{m}{}^{(I){}^{*}}\right]$ based on the fitness value. Extract the optimal one ${\hat{\theta }}^{*}{}_{I}$ and top $N^{*}$ individuals $D_{I}=$
$\left[{\hat{\theta }}_{1}{}^{(I){}^{*}};\;{\hat{\theta }}_{2}{}^{(I){}^{*}}\;;\ldots ;{\hat{\;\theta }}_{N{}^{*}}{}^{(I){}^{*}}\right]$ from $C_{I}$;(2) Modify the optimal individual ${\hat{\theta }}^{*}{}_{I}$. Calculate the center point (mean value) $\bar{\theta }=[{\bar{x}}_{1},\;{\bar{x}}_{2},\ldots ,{\bar{x}}_{n}{]}^{T}$ of $D_{I}$. Modify the optimal individual as, ${\hat{\theta }}^{*}{}_{I}=a{\hat{\theta }}^{*}{}_{I}+(1-a)\bar{\theta }$, a=rand(3) **If** the termination conditions is met The difference between two adjacent iterations error entropy is less than 0.0001, end the cycle.     **Else** $L_{I}{=\alpha L}_{I-1}$; Search for *m* individuals $B_{I}=\left[{\hat{\theta }}_{1}{}^{\left(I\right)};\;{\hat{\theta }}_{2}{}^{\left(I\right)};\ldots ;\;{\hat{\theta }}_{m}{}^{\left(I\right)}\right]$ within the neighborhood radius $L_{I}$ of ${\hat{\theta }}^{*}{}_{I}$; I=I+1;      **End**  **End**5. Get the estimation of parameters $\hat{\theta }$ and noise PDF $\hat{v}$.

After the coefficient $\hat{F}$ and noise ${\hat{v}}_{t}$ of the feedback-invariant are obtained, the benchmark entropy and output entropy could be obtained by Equations (8) and (11), and the CPA index could be obtained by Equation (9),
(36)$$\hat{\eta }=\frac{H(y_{t}{}^{mjc})}{H(y_{t})}=\frac{H^{\min}(\hat{F}{\hat{v}}_{t})}{H(y_{t})}$$

### 3.3. Algorithm Validation and Sensitivity Analysis of Initial Parameters

In this section, a model example is given to illustrate the initial parameters of improved algorithm, which further analyses the H-EDA and traditional EDA, consider the following system,
(37)$$y(k)=\frac{1+1.5z^{-1}+0.9z^{-2}}{1-1.7z^{-1}+0.7z^{-2}}v(k)$$

For the above example, the parameter vector to be identified is θ=[−1.7, 0.7, 1.5, 0.9]. Most of the initial parameters are based on the previous works [12,13,14,31,32,33]. Generally, in the EDA, the number of individuals in the initial population is N=1000; the optimal number of individuals for each iteration is m=200; and the maximum number of iterations is $n_{max}=120$. In WSA, the number of excellent individuals for crossover operation is $N{}^{*}=30$, R=800.

In this section, we focus on the sensitivity of shrinkage coefficient α and the initial value of neighborhood radius $L_{0}$. As mentioned before, the initial neighborhood radius is generally $L_{0}=(0.1\sim 0.3){\hat{\sigma }}_{v}$, and the shrinkage coefficient α is between 0.90 to 0.99. To find a proper set of parameters as much as possible, we have carried out a large number of experiments and some representative results are selected and attached in Table 1. In addition, we analyze the significance of a to the algorithm, a is a random number between [0, 1]. If a==1, it means that the crossover operation is not carried out. A novel parameter vector is defined as $P=\left[L_{0},\alpha ,a\right]$.

The test results show that the H-EDA is obviously superior to the traditional EDA. The global search capability of the algorithm is mainly determined by the neighborhood radius $L_{k}$. Too small a value of $L_{k}$ prompts the algorithm to get trapped in local optima, while too large of a value of $L_{k}$ slows down the algorithm; α can accelerate the later convergence of the algorithm, but too small of an α leads to rapid convergence and bad accuracy; the main function of a is to reduce iterations and prevent local optimality. In short, the setting of initial parameters must rely on a large number of experiments and the summary of previous works. For this test case, the optimal initial parameter vector can be selected as $P=[0.15{\hat{\sigma }}_{v},0.95,rand]$.

## 4. Simulation and Verification

To verify the effectiveness of the identification algorithm and the rationality of the performance assessment index, the following system is considered and the numerical simulation is carried out with Gaussian and non-Gaussian noise signals.
(38)$$y(t)=u(t-2)+\frac{1-0.2z^{-1}}{1-z^{-1}}v(t)$$

The transfer function of the controller is as follows,
(39)$$G_{c}=\frac{K}{1-0.2z^{-1}-0.8z^{-2}}$$

It is easy to know that the parameter vector of the system is θ=[−1,1,−1,−0.2], system delay τ=2, feedback-invariants F=[1,0.8], and the controller gain in the simulation is K=1.2. 

For more precise description, the follow-up content is divided into the following parts: in Section 4.1, the effect of the improved algorithm and the improved benchmark entropy on the performance evaluation are explained respectively when the expected output is unknown, and their effectiveness are verified respectively. In Section 4.2, the tracking index is used to evaluate the system performance when the expected output of the system is known.

### 4.1. Simulation When Expected Output Distribution Is Unknown

It can be concluded from Section 3 that the system parameter identification and noise PDF estimation should be carried out firstly when the expected output is unknown, then the benchmark value could be obtained according to the feedback invariant coefficient estimation $\hat{F}$ and noise estimation $\hat{v}$, and the performance evaluation index can be obtained finally. The specific process is shown in Figure 2.

According to the analysis ideas in Section 3.3, the parameters of the algorithms in this section are set as follows: 

In the EDA, the number of individuals in the initial population is N=1000, the optimal number of individuals for each iteration is m=200, and the maximum number of iterations is $n_{\max}=120$. Correspondingly, the parameters of H-EDA are set as, $N{}^{*}=30$, $L_{0}=0.15{\hat{\sigma }}_{v}$, α=0.95, R=800, the rests are consistent with the EDA. As the comparison algorithm, the parameters of the improved PSO (IPSO), particles N=50, acceleration factor $c_{1}=2$, $c_{2}=2$, $\;V_{max}=0.5$.

The termination condition is $|J^{\left(l\right)}-J^{\left(l-1\right)}|\leq 0.0001$ or reaches the specified number of iterations $n_{\max}$, $J^{\left(l\right)}$ is the l-generation GCL. 

The noises distribution are subject to normal N(0,0.255), Beta B(2,9), exponential E(0.5) and bimodal,
(40)v∼f(x)=a⋅rσ2π1e−(x−μ1)22σ12+b⋅1−rσ2π2e−(x−μ2)22σ22
where $\mu_{1}=-3$, $\mu_{2}=3$, $\sigma_{1}=1$, $\sigma_{2\;}=0.4$, r=0.7.

(1)When the fitness value is fixed as the GCL, compare the H-EDA with the traditional EDA and the IPSO.

Combine the above three algorithms with GCL, respectively. The probability distributions of the four kinds of noise are shown in Figure 3. It can be determined that the other three kinds of noise are non-Gaussian except for normal distribution. The results of the system parameter identification are shown in Table 2.

The results in Table 2 show that the H-EDA could estimate the parameters of CARMA model under different noises when the fitness value is determined, and the identification accuracy and speed are better. Moreover, the fitness value of the algorithm is the generalized correntropy loss function, which proves the effectiveness of the novel criterion in the identification process. It is evident that the noises distribution estimated by the H-EDA in Figure 4 and Figure 5 are very close to the real values. IPSO, as the comparison, is inferior to the H-EDA and cannot estimate the noise of bimodal distribution. Therefore, the superiority of the H-EDA is proved by the simulation results.

According to the noise PDF ${\hat{v}}_{t}$ and the feedback invariant coefficient $\hat{F}$, we can get the performance assessment indexes, as shown in Table 3, where $\eta_{ME}$ represents the theoretical benchmark index; ${\hat{\eta }}_{MJ-EDA}$ is the index obtained by identification results of the traditional EDA based on GCL; ${\hat{\eta }}_{MJ-H-EDA}$ corresponds to the H-EDA index. Obviously, ${\hat{\eta }}_{MJ-H-EDA}$ is closer to the theoretical value, which proves the effectiveness of the hybrid-algorithm again.

(2)When the identification algorithm is fixed as the H-EDA, compare the traditional correntropy (TC) with the generalized correntropy criterion (GCL).

In the previous section, we have proved the effectiveness of generalized correntropy criterion in the performance assessment process. Next, we compare the superiority of GCL with TC based on the H-EDA. The parameters of algorithms, identified objects and noise PDFs are consistent with (1). The simulation results are illustrated in Table 4 and Figure 6. 

The results show that when the disturbance obeys Gaussian, non-Gaussian and bimodal distribution, the identification results based on GCL are more accurate, and the time required to obtain the evaluation index is shorter.

### 4.2. Simulation When the Expected Output Distribution Is Known

Assumed that the known expected output distribution is Gaussian with mean value of 0 and variance of 1, as described in Section 2, the output PDF is expected to track the target PDF as much as possible.

The system under Gaussian and non-Gaussian noise disturbance are simulated respectively when the expected output distribution of the system is known. The output PDF tracks the input PDF as shown in Figure 7. The noise in (a) obeys normal distribution, while the noise in (b) obeys the non-Gaussian distribution. The generalized correntropy performance assessment index could be calculated by Formula (18) and shown in Table 5. 

In Table 5, $\eta_{GC}$ is the generalized correntropy index and $\eta_{C}$ is the traditional correntropy index. Assessment results of generalized correntropy are consistent with the traditional correntropy index when the controller gain *K* is different, which also proves the effectiveness of the new index.

## 5. Conclusions

A new CPA method for linear non-Gaussian systems based on generalized correntropy is proposed by analyzing the limitations of existing performance evaluation methods and indicators. The simple tracking index based on generalized correntropy is given directly when the expected output is known. When the expected output is unknown, the generalized correntropy loss function is used as the fitness value of the H-EDA to estimate the system parameters and noise distribution, so as to calculate the system performance benchmark more quickly and accurately, and then the performance assessment results can be obtained. The single variable method is used to simulate the SISO control system under different noise disturbances, and the effectiveness of the new index and H-EDA is verified respectively. The results show that generalized correntropy index is superior to the traditional correntropy in evaluating the control loop performance of linear systems with non-Gaussian stochastic disturbances, and the H-EDA could also identify the system parameters more quickly and accurately. Then, the follow-up research still focuses on the selection of the new performance assessment index and the improvement of identification algorithm.

## Figures and Tables

**Figure 1 entropy-23-00764-f001:**
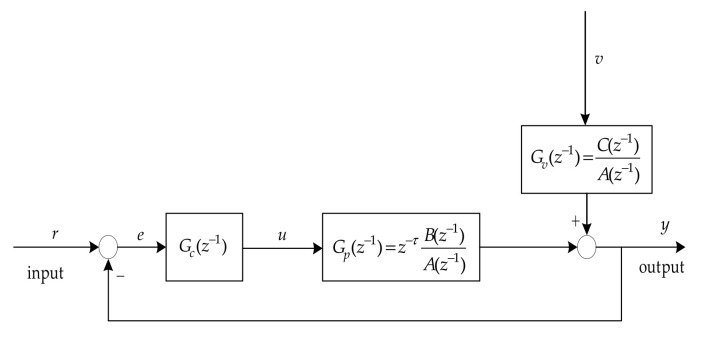
SISO feedback control loop system.

**Figure 2 entropy-23-00764-f002:**
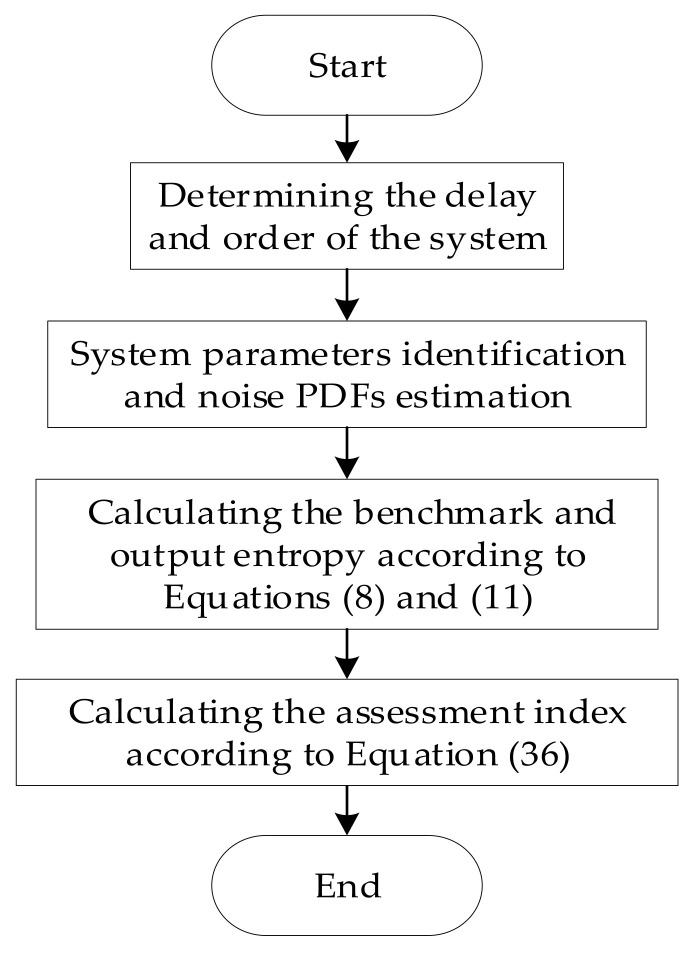
Flow chart of performance assessment.

**Figure 3 entropy-23-00764-f003:**
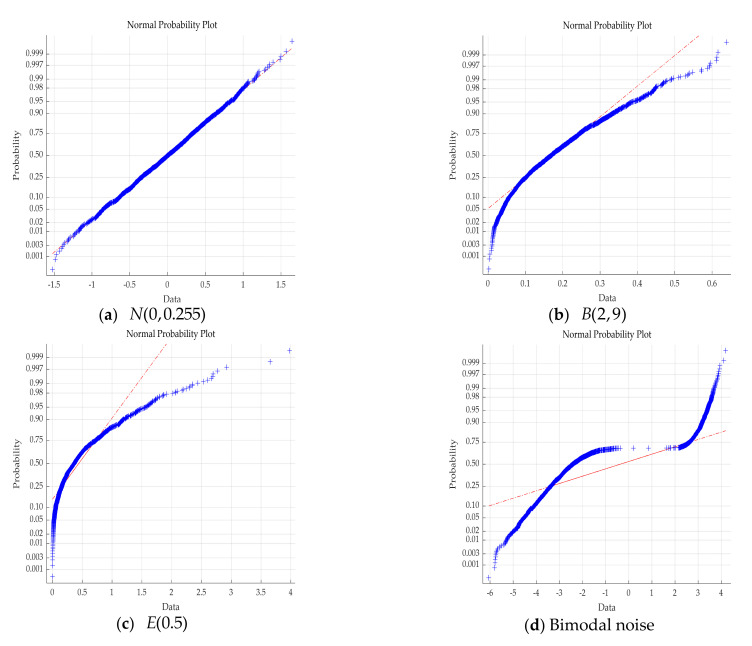
Probability distribution of different noises.

**Figure 4 entropy-23-00764-f004:**
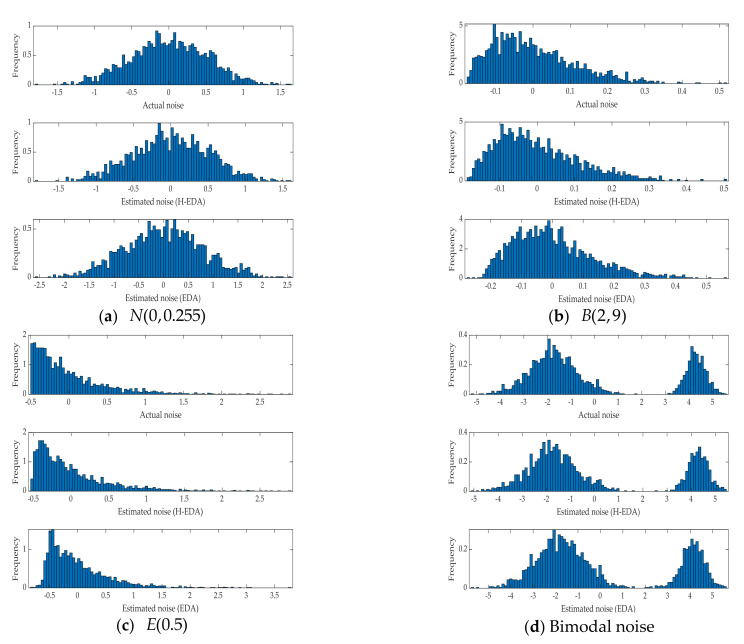
Actual and estimated probability distribution histograms of different noises.

**Figure 5 entropy-23-00764-f005:**
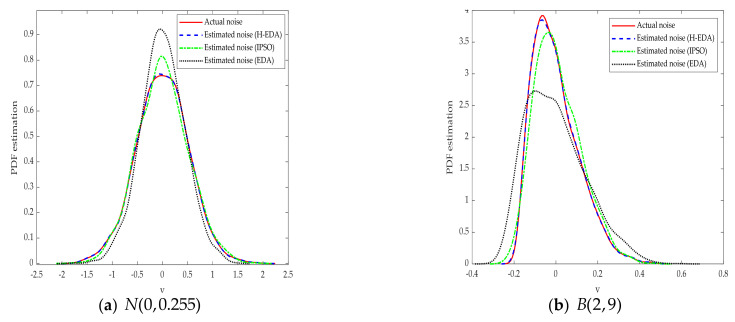
Actual and estimated PDF of different noises.

**Figure 6 entropy-23-00764-f006:**
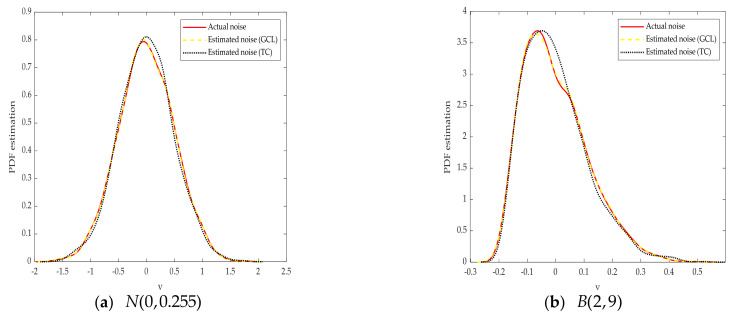
Actual and estimated PDF with different noises.

**Figure 7 entropy-23-00764-f007:**
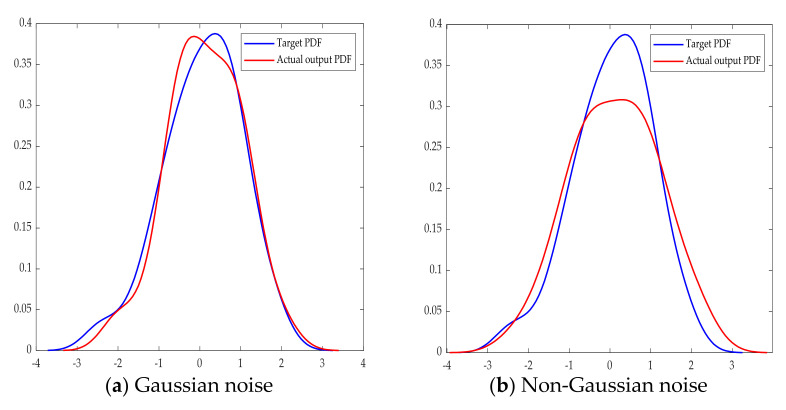
Expected and actual output PDF under different noises.

**Table 1 entropy-23-00764-t001:** Model identification results based on the parameters sensitivity analysis (partial).

Project	Parameter Configuration			Identification Results	Iterations
EDA	N=1000 , m=200			[−1.4922, 0.5527, 1.6509, 0.6267]	120
H-EDA	$L_{0}=0.1{\hat{\sigma }}_{v}$	α=0.93	a=rand	[−1.7209, 0.7241, 1.4680, 0.8668]	75
			a==1	[−1.7398, 0.7529, 1.4099, 0.7911]	105
		α=0.95	a=rand	[−1.7309, 0.7227, 1.4219, 0.8113]	82
			a==1	[−1.6675, 0.6728, 1.4003, 0.7849]	109
		α=0.97	a=rand	[−1.7273, 0.7305, 1.4085, 0.8173]	91
			a==1	[−1.6631, 0.6645, 1.4204, 0.8054]	120
	$L_{0}=0.15{\hat{\sigma }}_{v}$	α=0.93	a=rand	[−1.7197, 0.7218, 1.5160, 0.8851]	76
			a==1	[−1.7749, 0.7735, 1.4595, 0.8299]	87
		α=0.95	a=rand	[−1.6919, 0.6975, 1.5009, 0.9104]	62
			a==1	[−1.7181, 0.7184, 1.5090, 0.8544]	79
		α=0.97	a=rand	[−1.7304, 0.7310, 1.5164, 0.8964]	95
			a==1	[−1.6643, 0.6654, 1.4698, 0.8605]	113
	$L_{0}=0.2{\hat{\sigma }}_{v}$	α=0.93	a=rand	[−1.7112, 0.7176, 1.3935, 0.8291]	71
			a==1	[−1.7672, 0.7689, 1.1863, 0.5712]	97
		α=0.95	a=rand	[−1.6734, 0.6737, 1.5061, 0.9069]	104
			a==1	[−1.7462, 0.7504, 1.4771, 0.8579]	117
		α=0.97	a=rand	[−1.6823, 0.7446, 1.4500, 0.8403]	109
			a==1	[−1.7769, 0.7789, 1.4058, 0.8304]	120

**Table 2 entropy-23-00764-t002:** Identification results of two algorithms under different noise distributions.

	**Actual Value**	***N* (0, 0.255)**		***B* (2, 9)**	
		**EDA**	**H-EDA**	**EDA**	**H-EDA**
$a_{1}$	−1	−0.9342	−0.9873	−1.0529	−1.0027
$b_{0}$	1	0.9186	0.9968	1.0848	1.0003
$b_{1}$	−1	−0.9337	−0.9816	−0.9466	−0.9992
$c_{1}$	−0.2	−0.0879	−0.1991	−0.2365	−0.2014
F	[1, 0.8]	[1, 0.7011]	[1, 0.7782]	[1, 0.8596]	[1, 0.8035]
Time(s)		29.9517	15.2724	35.4492	13.6584
		***E* (0.5)**		**Bimodal Noise**	
		**EDA**	**H-EDA**	**EDA**	**H-EDA**
$a_{1}$	−1	−0.9158	−0.9884	−0.9173	−1.0097
$b_{0}$	1	0.9389	0.9961	0.9377	1.0643
$b_{1}$	−1	−0.9323	−0.9924	−0.8946	−1.0853
$c_{1}$	−0.2	−0.0865	−0.1935	−0.1556	−0.2037
F	[1, 0.8]	[1, 0.7595]	[1, 0.7857]	[1, 0.6987]	[1, 0.8138]
Time(s)		30.9885	14.2616	29.6255	14.7962

**Table 3 entropy-23-00764-t003:** CPA indexes of two comparison algorithms.

Index	*N* (0, 0.255)	*B* (2, 9)	*E* (0.5)	Bimodal Noise
$\eta_{ME}$	0.8809	0.9357	0.9408	0.9332
${\hat{\eta }}_{MJ-EDA}$	0.8345	0.9016	09134	0.9267
${\hat{\eta }}_{MJ-H-EDA}$	0.8794	0.9388	0.9399	0.9336

**Table 4 entropy-23-00764-t004:** Estimated parameters and evaluation indexes under two different entropy benchmarks.

	**Actual** **Value**	***N* (0, 0.255)**		***B* (2, 9)**	
		**Results of Traditional Correntropy**	**Results of Generalized Correntropy**	**Results of Traditional Correntropy**	**Results of Generalized Correntropy**
$a_{1}$	−1	−0.9312	−0.9873	−1.0148	−1.0027
$b_{0}$	1	1.0061	0.9968	1.0254	1.0003
$b_{1}$	−1	−1.1073	−0.9816	−1.1995	−0.9992
$c_{1}$	−0.2	−0.1635	−0.1991	−0.1837	−0.2014
F	[1, 0.8]	[1, 0.7149]	[1, 0.7782]	[1, 0.8434]	[1, 0.8035]
$\hat{\eta }$		0.8566	0.8794	0.8927	0.9388
Time(s)		18.9443	15.2724	15.1667	13.6584
		***E* (0.5)**		**Bimodal Noise**	
$a_{1}$	−1	−0.9838	−0.9884	−0.9945	−1.0097
$b_{0}$	1	0.9246	0.9961	1.0852	1.0643
$b_{1}$	−1	−0.9697	−0.9924	−1.0521	−1.0853
$c_{1}$	−0.2	−0.2194	−0.1935	−0.1798	−0.2037
F	[1, 0.8]	[1, 0.7520]	[1, 0.7857]	[1, 0.8102]	[1, 0.8138]
$\hat{\eta }$		0.9071	0.9399	0.9462	0.9336
Time(s)		18.9936	14.2616	18.5615	14.7962

**Table 5 entropy-23-00764-t005:** Generalized correntropy performance index under expected output distribution is known.

	Index	Gaussian Noise	Non-Gaussian Noise
*K* = 0.8	$\eta_{C}$	0.8406	0.6554
	$\eta_{GC}$	0.8215	0.6021
*K* = 1.0	$\eta_{C}$	0.8698	0.7077
	$\eta_{GC}$	0.8520	0.6641
*K* = 1.2	$\eta_{C}$	0.8874	0.7447
	$\eta_{GC}$	0.8162	0.6429

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
