# Peer review of "Generalized Correntropy Criterion-Based Performance Assessment for Non-Gaussian Stochastic Systems"

_entropy, 2021, doi:10.3390/e23060764_

Round 1

Reviewer 1 Report

The paper focuses on the CPA (control loop performance assessment), while the proposed evaluation method is based on the GCC (generalized correntropy criterion), which is usually related to robustness features. Thus, the method is suitable for non-Gaussian stochastic systems. The method is also based on some existing techniques, like EDA, WSA, RELS, etc. From this point of view, the contribution is incremental and the novelty is limited. Several comments and suggestions are provided in the following.

1) There is a strong similarity between the control loop systems (as shown in Figure 1) and the adaptive filtering configurations. In addition, GCC was also used in the context of adaptive filters, targeting improved robustness to impulsive noises, heavy-tail distributions, etc. The authors could mention this aspect, also indicating several appropriate references.

2) Section 3.2.1 could be extended. It is not very clear how the parameter \theta results based on the RELS algorithm.

3) The shrinkage coefficient \alpha is chosen as 0.95 (in Section 3.2.2). The authors could provide more details/arguments about this choice, especially since the search strategy presented in this section is of great importance.

4) Similar issue related to the parameter a in (28). How sensitive is the solution to the choice of this parameter?

5) The arguments/explanations provided in the end of Section 3.2.3 sound quite vague and should be detailed. What is the significance of an “excellent solution” and how it can be assess? Same issue in Section 3.2.5.

6) The results in Figure 2 could be depicted using a log scale, for a better readability, for example, between EDA and H-EDA.

7) There are also some editorial issues that should be addressed. For example:

- Please check that all the acronyms are defined within the paper. For example, the acronym SISO should be defined in the end of Section 1. Similar issue related to the CARMA acronym. All the acronyms should be defined in the paper (at the first appearance), even if some of them are well-known.

- In many places within the paper, there are mentions about “EDA algorithm”. In fact, the acronym EDA already contains the word algorithm (EDA = estimation of distribution algorithm), so it should be mentioned only as EDA. Similar issue related to WSA, which also contains the word “algorithm” (WSA = Wading across the stream algorithm), so we should never say “WSA algorithm”.

- It is recommended to use different line types in Figures 2, 6, 7, and 8.

- An appropriate legend should be included in Figure 4.

- Some sentences should be reformulated / clarified. For example:

* before eq. (9):

?? “we can get the CPA index based on MEC is”

* after eq. (9):

?? “indicating the system is approach to optimal”

* end of the paragraph that follows eq. (11):

?? “As shown in Equation (9).”

... etc.

- Avoid using short forms (contractions) in scientific papers. For example:

correntropy doesn’t have --> correntropy does not have

it's an extension --> it is an extension

SE can’t be used  --> SE cannot be used

... etc.

- Please check for typos. For example, in Section 4 (on pages 16 and 18):

corrntropy --> correntropy

Author Response

Response to Reviewer 1 Comments

Thank you for your carefulness and sense of responsibility. These valuable suggestions will make our article more rigorous and scientific. For the questions you have proposed, I have made a detailed response in this document.

Point 1: There is a strong similarity between the control loop systems (as shown in Figure 1) and the adaptive filtering configurations. In addition, GCC was also used in the context of adaptive filters, targeting improved robustness to impulsive noises, heavy-tail distributions, etc. The authors could mention this aspect, also indicating several appropriate references.

Response 1: Thank you very much. This suggestion is very important for our article. As you said, the inspiration of generalized correntropy comes from its application in robust adaptive filtering. We have supplemented some relevant literature to better illustrate the wide application of generalized correntropy in non-Gaussian stochastic systems. The references added include [17-20], they are all supplemented in the introduction.

Point 2: Section 3.2.1 could be extended. It is not very clear how the parameter results based on the RELS algorithm.

Response 2: RELS is a classical parameter identification algorithm, which is often used to identify the parameters of the ARMA model. We extend it in Section 3.2.1 and give a more detailed description. The added formulas include (23), (24), (25), (26), they can clearly explain how the parameter is obtained.

Point 3: The shrinkage coefficient is chosen as 0.95 (in Section 3.2.2). The authors could provide more details/arguments about this choice, especially since the search strategy presented in this section is of great importance.

Response 3: This suggestion is very valuable for our algorithm verification in this paper, thank you again. The shrinkage coefficient is set to accelerate the later convergence of the algorithm. The smaller is, the faster the convergence is. However,  should not be too small, otherwise, the algorithm will converge rapidly and fall into local optimum. Therefore, the value α is generally between 0.9 and 0.99, which should be adjusted flexibly according to different model systems. To illustrate the sensitivity of parameters systematically, we add a numerical model (37) in Section 3.3. To find a proper set of parameters as much as possible, we have carried out a large number of experiments based on the model (37). Then, some representative results are selected and attached in Table 1. We can choose the optimal parameter setting for the current model (37) by analyzing the results in Table 1.

Point 4: Similar issue related to the parameter a in (28). How sensitive is the solution to the choice of this parameter?

Response 4: The crossover operation is set to avoid the occasionality of a single individual. It is inspired by the genetic algorithm, which can retain the information of excellent individuals to the greatest extent. Parameter a is a random number between 0 and 1, so it is not necessary to analyze how sensitive the solution is to the choice of this parameter specifically. We are more concerned about its positive significance for the algorithm. The detailed analysis process is given in Section 3.3. If a==1, it means that the crossover operation is not carried out. Similarly, we have carried out a lot of experiments. The results in Table 1 can reflect the positive role of a in reducing iterations of the algorithm.

The setting of initial parameters is not a simple problem, it must rely on a large number of experiments and the summary of previous works.

Point 5: The arguments/explanations provided at the end of 3.2.3 sound quite vague and should be detailed. What is the significance of an “excellent solution” and how it can be assessed? Same issue in Section 3.2.5.

Response 5: Aiming at the problems in Section 3.2.3, we give a more detailed illustration in (30). Excellent solutions represent the individuals with the top fitness value. In each iteration, the algorithm search for m individuals around the optimal solution. Calculate the fitness value of these individuals and sort them based on the fitness value. The updated optimal solution is the first individual with the best fitness, and the excellent solutions are the top individuals. In Section 3.2.5, we provide very detailed algorithm program steps.

Point 6: The results in Figure 2 could be depicted using a log scale, for better readability, for example, between EDA and H-EDA.

Response 6: To analyze the sensitivity of algorithm parameters more systematically, we have to delete the original test example, and replace it with a more suitable numerical model (37) and describe it in more detail. The superiority of H-EDA can also be proved by the results in Section 3.3.

Point 7: There are also some editorial issues that should be addressed.

Response 7: Thank you for your preciseness and dedication. As for the editing problems, we have checked and revised the whole article. Including the typos you mentioned, ambiguous statements, and the line patterns in Figures 2, 6, and 7.

Reviewer 2 Report

This work integrates the idea of Wading across the stream algorithm (WSA) into the traditional EDA to improve the local convergence ability of the algorithm. 
Simulation of a SISO system was conducted in this work. 
Test results show the identification results of the generalized correntropy criterion are more accurate, and the time required to obtain the evaluation index is shorter.

Some more detailed comments are given below. I hope that if the authors will take them into account the paper will be improved.
1. Except Ref. [12]、[20]、[29], it has almost no recent reference from the last two years. Recent and state-of-the-art references should be cited on the topic.
2. There are many journal paper concerning the estimation of distribution algorithm from the last two years.
Authors should survey and discuss these state-of-the-art in Section 3.
3. Except the traditional EDA, authors should compare the proposed H-EDA with some state-of-the-art methods for test example in Section 4.
4. Provide all parameter setting of the traditional EDA and H-EDA in section 4.
5. Authors should add a new subsection in Section 3.2 to discuss how to find a set of proper initial parameters of the proposed method? It seems to me that it is not an easy job and is really problem dependent to select the parameters, since there are lots of parameters influencing the performance of the proposed H-EDA together. 
6.  It is hard to link the proposed H-EDA with the test example.
A simple numerical example is needed to provide an illustration between the traditional EDA and H-EDA in Section 3.

Author Response

Response to Reviewer 2 Comments

Thank you for your carefulness and sense of responsibility. These valuable suggestions will make our article more rigorous and scientific. For the questions you have proposed, I have made a detailed response in this document.

Point 1: Except Ref. [12], [20], [29], it has almost no recent reference from the last two years. Recent and state-of-the-art references should be cited on the topic.

Response 1: Thank you very much. This suggestion is very important for our article. As you mentioned, the literature cited in the previous article is not state-of-the-art enough, so we have made up for it. We have cited and analyzed the latest literature in terms of performance assessment [12, 14, 33], improvement of EDA [27-30], and the application of generalized correntropy [17-20].

Point 2: There are many journal paper concerning the estimation of distribution algorithm from the last two years. Authors should survey and discuss these state-of-the-art in Section 3.

Response 2: EDA algorithm has made great development and application in recent years, which is one of the reasons why I choose it. In Section 3.1, we discuss several latest EDA examples [27-30] and further realize their potential. For example, Wang [28] proposed a hybrid algorithm named estimation of particle swarm distribution algorithm (EPSDA), which combines PSO (the local search method) with EDA (the global search method) to improve the efficiency of solving nonlinear bilevel programming problems (NBLP). This is an ingenious improvement ideal, which can solve the problem of poor local search capability in the EDA algorithm, so as WSA.

Point 3: Except the traditional EDA, authors should compare the proposed H-EDA with some state-of-the-art methods for test example in Section 4.

Response 3: In the process of CPA, EDA is mainly used to identify the parameters and estimate the PDFs of the given system. Previous literature only used the traditional EDA based on entropy criterion, and only to replace the fitness value of the algorithm. In this paper, the hybrid EDA is applied to the performance assessment process for the first time. The new fitness value function (GCL) can also describe the statistical characteristics of non-Gaussian stochastic variables more comprehensively, which greatly improves the efficiency of parameter identification.

To further illustrate the superiority of H-EDA, we set the improved particle swarm optimization (IPSO) algorithm as the control group. The simulation results of several algorithms are shown in Figure 6 of Section 4.1 (1). For the first three kinds of noise, the noise PDFs can be estimated successfully by all several algorithms, but the IPSO algorithm is invalid under the bimodal distribution. The performance of H-EDA is superior to other algorithms under four kinds of noise, which can prove the effectiveness of H-EDA in the process of performance evaluation.

Of course, due to the limitation of time and team level, we are still unable to apply the most advanced PSO algorithm. In future researches, we will apply more state-of-the-art algorithms to the performance evaluation process.

Point 4: Provide all parameter setting of the traditional EDA and H-EDA in section 4.

Response 4: In Section 4, we add all the missing parameters of EDA and H-EDA. 

Point 5: Authors should add a new subsection in Section 3.2 to discuss how to find a set of proper initial parameters of the proposed method? It seems to me that it is not an easy job and is really problem dependent to select the parameters, since there are lots of parameters influencing the performance of the proposed H-EDA together.

Point 6: It is hard to link the proposed H-EDA with the test example.

A simple numerical example is needed to provide an illustration between the traditional EDA and H-EDA in Section 3.

Response 5, 6: This suggestion is very valuable for our algorithm verification in this paper, thank you again. In this paper, most of the initial parameters are based on the previous works [12, 13,14, 31, 32, 33]. We focus on the sensitivity of shrinkage coefficient α and the initial value of neighborhood radius L.  In addition, we analyze the significance of a to the algorithm.

The local search ability of the algorithm is mainly determined by the neighborhood radius Lk. Too small Lk prompts the algorithm to get trapped in local optima, while too large Lk slows down the algorithm. The shrinkage coefficient α is set to accelerate the later convergence of the algorithm. The smaller α is, the faster the convergence is. However, α should not be too small, otherwise, the algorithm will converge rapidly and fall into local optimum. Therefore, the value of α is generally between 0.9 and 0.99, which should be adjusted flexibly according to different model systems. To illustrate the sensitivity of parameters systematically, we add a numerical model (37) in Section 3.3. To find a proper set of parameters as much as possible, we have carried out a large number of experiments based on the model (37). Then, some representative results are selected and attached in Table 1. We can select the optimal parameter setting for the current model (37) by analyzing the results in Table 1.

The crossover operation is set to avoid the occasionality of a single individual. It is inspired by the genetic algorithm, which can retain the information of excellent individuals to the greatest extent. Parameter a is a random number between 0 and 1, so it is not necessary to analyze how sensitive the solution is to the choice of this parameter specifically. We are more concerned about its positive significance for the algorithm. The detailed analysis process is given in Section 3.3. If a==1, it means that the crossover operation is not carried out. Similarly, we have carried out a lot of experiments. The results in Table 1 can reflect the positive role of a in reducing iterations of the algorithm. The superiority of H-EDA can also be proved by the results in Section 3.3.

The setting of initial parameters is not a simple problem, it must rely on a large number of experiments and the summary of previous works.

Round 2

Reviewer 1 Report

The authors have properly addressed my comments.

I would recommend a final proofreading. For example:

- on page 3, line 120:

next scetion --> next section

- on page 5, line 187:

will to be repeated --> will be repeated

etc.

- on page 5, “ln” is used in eq. (10) and “log” in eq. (11)

- on page 11, line 376:

\alpha can accelerates --> \alpha can accelerate

- on page 12, line 398, in the title of subsection 4.1:

When --> when

- on page 13, line 414:

reachs --> reaches

- on page 17, line 455, in the title of subsection 4.2:

When The --> when the

- on page 17, line 521:

IEEE Transactions on Systems --> IEEE Transactions on Systems, Man, and Cybernetics: Systems

Author Response

Thank you for your responsible comments. We have checked the whole paper and revised the suspicious content. For Equations (10) and (11), Ln is just the special form of a log, which represents the logarithmic function with a base value of 10. Thank you for your rigorous and serious attitude towards academic work. Best wishes to you.

Reviewer 2 Report

The authors have carefully addressed the previous comments of the reviewer and significantly improved the manuscript. 

Author Response

Thank you for your responsible comments. We have checked the whole paper and revised the suspicious content.  Thank you for your rigorous and serious attitude towards academic work. Best wishes to you.

This manuscript is a resubmission of an earlier submission. The following is a list of the peer review reports and author responses from that submission.